# THE MIRAGE OF ACTION-DEPENDENT BASELINES IN REINFORCEMENT LEARNING

**George Tucker**[1], **Surya Bhupatiraju**[1]*, **Shixiang Gu**[1,2,3], **Richard E. Turner**[2], **Zoubin Ghahramani**[2,5], **& Sergey Levine**[1,4]

{gjt,sbhupatiraju}@google.com,{sg717,ret26}@cam.ac.uk, zoubin@eng.cam.ac.uk,svlevine@eecs.berkeley.edu

[1]Google Brain, USA
[2]University of Cambridge, UK
[3]Max Planck Institute for Intelligent Systems, Tubingen, Germany
[4]UC Berkeley, USA
[5]Uber AI Labs, USA

## ABSTRACT

Model-free reinforcement learning with flexible function approximators has shown success in goal-directed sequential decision-making problems. Policy gradient methods are a widely used class of stable model-free algorithms and typically, a state-dependent baseline or control variate is necessary to reduce the gradient estimator variance. Several recent papers extend the baseline to depend on both the state and action, and suggest that this enables significant variance reduction and improved sample efficiency without introducing bias into the gradient estimates. To better understand this development, we decompose the variance of the policy gradient estimator and numerically show that learned state-action-dependent baselines do not in fact reduce variance over a state-dependent baseline in the commonly tested benchmark domains. We confirm this unexpected result by reviewing the open-source code accompanying these prior papers, and show that subtle implementation decisions cause deviations from the methods presented in the papers and explain the sources of the previously observed empirical gains.

## 1 INTRODUCTION

Policy gradient methods (Williams, 1992; Sutton et al., 2000; Kakade, 2002; Peters & Schaal, 2006; Silver et al., 2014; Schulman et al., 2015a; 2017) are a class of model-free RL algorithms that have found widespread adoption due to their stability and ease of use. Gu et al. (2017b); Grathwohl et al. (2018); Liu et al. (2018); Wu et al. (2018) present promising results extending the classic state-dependent baselines to state-action-dependent baselines. This line of investigation is attractive, because baselines do not introduce bias and thus do not compromise the stability of the underlying policy gradient algorithm. We present a decomposition of the variance of the policy gradient estimator, which isolates the potential variance reduction due to state-action-dependent baselines. We numerically evaluate the variance components on benchmark continuous control tasks and draw two conclusions: (1) on these tasks, a learned state-action-dependent baseline does not significantly reduce variance over a learned state-dependent baseline, a conclusion seemingly at odds with previous work, and (2) the variance caused by using a function approximator for the value function or state-dependent baseline is much larger than the variance reduction from adding action dependence to the baseline.

## 2 POLICY GRADIENT VARIANCE DECOMPOSITION

The policy gradient estimator typically suffers from high variance. Several recent methods (Gu et al., 2017b; Thomas & Brunskill, 2017; Grathwohl et al., 2018; Liu et al., 2018; Wu et al., 2018)

---

*Work done while the author participated in the Google AI Residency.

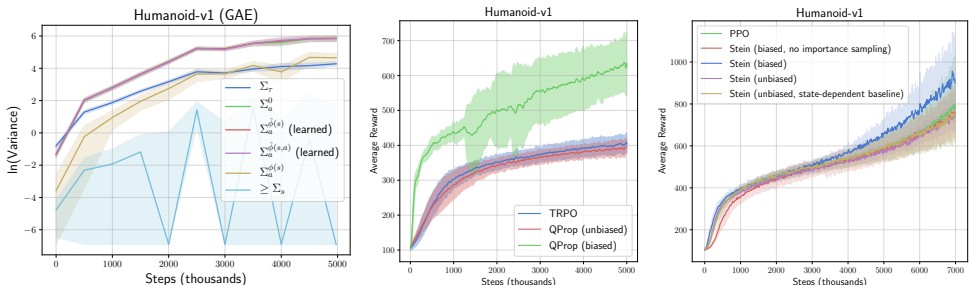

Figure 1: (Left) Evaluating the variance terms (Eq. 2) of the gradient estimator on Humanoid. The "learned" label in the legend indicates that a function approximator to $\phi$ was used. Note that when using $\phi(s,a) = \hat{A}(s,a)$, $\Sigma_a^{\hat{\phi}(s,a)}$ is 0, so is not plotted. The upper and lower bands indicate two standard errors of the mean. (Middle) Evaluation of the implementation of Q-Prop from Gu et al. (2017b), an unbiased implementation of Q-Prop that applies the normalization to all terms, and TRPO. (Right) Evaluation of the implementation of the Stein control variate and PPO from Liu et al. (2018), a biased variant of Stein without importance sampling, an unbiased variant of Stein, and an unbiased variant of Stein using only a state-dependent baseline. See Appendix Section 7 for details.

have extended baselines (Williams, 1992; Weaver & Tao, 2001) to depend on the action as well as the state. With a state-action-dependent baseline $\phi(s,a)$, the policy gradient estimator is

$$\hat{g} = \left( \hat{A}(s,a,\tau) - \phi(s,a) \right) \nabla \log \pi(a|s) + \nabla \mathbb{E}_{a \sim \pi} \left[ \phi(s,a) \right], \tag{1}$$

where $\pi(a|s)$ is the policy's distribution over actions, $\tau$ is the rest of the trajectory, $\hat{A}$ is an estimator of the advantage function, and $\nabla \mathbb{E}_{a \sim \pi} \left[ \phi(s,a) \right]$ is either analytically evaluated (Gu et al., 2017b) or estimated with the reparameterization trick (Liu et al., 2018; Grathwohl et al., 2018) (See Appendix Section 5 for background information). The variance[1] of the policy gradient estimator, $\Sigma := \text{Var}_{s,a,\tau}(\hat{g})$, can be decomposed as (see Apendix Section 6 for derivation)

$$\Sigma = \underbrace{\mathbb{E}_{s,a} \left[ \text{Var}_{\tau|s,a} \left( \hat{A}(s,a,\tau) \nabla \log \pi(a|s) \right) \right]}_{\Sigma_\tau} + \underbrace{\mathbb{E}_s \left[ \text{Var}_{a|s} \left( \left( \hat{A}(s,a) - \phi(s,a) \right) \nabla \log \pi(a|s) \right) \right]}_{\Sigma_a}$$
$$+ \underbrace{\text{Var}_s \left( \mathbb{E}_{a|s} \left[ \hat{A}(s,a) \nabla \log \pi(a|s) \right] \right)}_{\Sigma_s}. \tag{2}$$

We estimate the magnitude of the three terms for benchmark continuous action tasks as training proceeds. Once we decide on the form of $\phi(s,a)$, approximating $\phi$ is a learning problem in itself. To understand the approximation error, we evaluate the situation where we have access to an oracle $\phi(s,a)$ and when we learn a function approximator for $\phi(s,a)$. We plot each of the individual terms $\Sigma_\tau, \Sigma_a$, and $\Sigma_s$ of the gradient estimator variance for Humanoid in Figure 1 (and HalfCheetah in Appendix Figure 3).

We plot the variance decomposition for two choices of $\hat{A}(s,a,\tau)$: the discounted return, $\sum_t \gamma^t r_t$ (Appendix Figure 2), and GAE Schulman et al. (2015b). In both cases, we set $\phi(s) = \mathbb{E}_{a \sim \pi} \left[ \hat{A}(s,a) \right]$ and $\phi(s,a) = \hat{A}(s,a)$ (the optimal state-action-dependent baseline). When using the discounted return, we found that $\Sigma_\tau$ dominates $\Sigma_a^{\phi(s)}$, suggesting that even an optimal state-action-dependent baseline (which would reduce $\Sigma_a$ to 0) would not improve over a state-dependent baseline (Appendix Figure 2). When using GAE, $\Sigma_\tau$ is reduced and now the optimal state-dependent baseline could theoretically improve performance over a state-dependent baseline. However, when we used neural network function approximators to estimate $\phi$, we found that the state-dependent and state-action-dependent function approximators produced similar, but much higher variance than when using an oracle $\phi$. This suggests that, in practice, we would not see improved performance using a state-action-dependent baseline over a state-dependent baseline on these tasks.

---

[1]We assume we can analytically evaluate the expectation over $a$ in the second term because it only depends on $\phi$ and $\pi$, which we can evaluate multiple times without explicitly querying the environment.

Furthermore, we see that closing the function approximation gap of $V(s)$ and $\phi(s)$ would produce much larger reductions in variance than from using the optimal state-action-dependent baseline over the state-dependent baseline. This suggests that improved function approximation of both $V(s)$ and $\phi(s)$ should be a priority.

## 3 UNVEILING THE MIRAGE

This appears to be a paradox: if the variance due to the action distribution is not reduced, how are prior methods that propose state-action-dependent baselines able to report significant improvements in learning performance? We analyze implementations accompanying these works (Gu et al., 2017b; Liu et al., 2018; Grathwohl et al., 2018), and show that they actually introduce bias into the policy gradient due to subtle implementation decisions. We find these methods are effective not because of unbiased variance reduction, but instead because they trade bias for variance.

### 3.1 ADVANTAGE NORMALIZATION

Although Q-Prop and IPG (Gu et al., 2017a) (when $\nu = 0$) claim to be unbiased, the implementations of Q-Prop and IPG apply an adaptive normalization to only some of the estimator terms, which introduces a bias. We found that the adapative bias-variance trade-off induced by the asymmetric normalization is crucial for the gains observed in Gu et al. (2017b). If implemented as unbiased, it does not outperform TRPO (Figure 1).

### 3.2 POORLY FIT VALUE FUNCTIONS

The GAE advantage estimator has mean zero when $\hat{V}(s) = V^\pi(s)$, which suggests that a state-dependent baseline is unnecessary if $\hat{V}(s) \approx V^\pi(s)$. However, when $\hat{V}(s)$ poorly approximates $V^\pi(s)$, the GAE advantage estimator has nonzero mean, and a state-dependent baseline can reduce variance. We show that is the case by taking the open-source code accompanying Liu et al. (2018), and implementing a state-dependent baseline. It achieves comparable variance reduction to the state-action-dependent baseline (Appendix Figure 6).

### 3.3 SAMPLE-REUSE IN BASELINE FITTING

Fitting the baseline to the current batch of data and then reusing the data and updated baseline to form the policy gradient estimator results in a biased gradient estimate (Jie & Abbeel, 2010). On Humanoid, the biased method implemented in Liu et al. (2018) performs best (Figure 1), and on HalfCheetah, the biased methods suffer from instability and underperforms an unbiased variant (Appendix Figure 5). The same considerations apply to fitting the value function (Jie & Abbeel, 2010; Schulman et al., 2015b). Grathwohl et al. (2018) fit the value function before estimating the policy gradient step, introducing bias into their estimator. After correcting these issues in their implementation, we do not observe an improvement (Appendix Figure 7).

## 4 DISCUSSION

State-action-dependent baseline promise variance reduction without introducing bias. We clarify the practical effect of state-action-dependent baselines in common continuous control benchmark tasks. In practice, currently used function approximators are unable to achieve significant variance reduction. Furthermore, we found that much larger variance reduction might be achieved by instead improving the accuracy of the value function or the state-dependent baseline function approximators. We re-examined previous work on state-action-dependent baselines and identified a number of pitfalls. We also were able to correctly attribute the empirical results previously observed to implementation decisions that introduce bias in exchange for variance reduction.

Finally, we note that the relative contributions of each of the terms to the policy gradient variance are problem specific. We evaluated the terms on continuous control benchmarks, however, we expect that in some environments, $\Sigma_a$ may be much larger, such as in a discrete task with a critical decision point. In these cases, we see practical gains from using a state-action-dependent baseline (Wu et al., 2018; Grathwohl et al., 2018).

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

APPENDIX

## 5 BACKGROUND

Reinforcement learning aims to learn a policy for an agent to maximize a sum of reward signals (Sutton & Barto, 1998). To begin, the agent starts at an initial state $s_0 \sim P(s_0)$. The agent then repeatedly samples an action $a_t$ according to a parameterized policy $\pi_\theta(a_t|s_t)$, receives a reward $r_t \sim P(r_t|s_t, a_t)$ and transitions to a subsequent state $s_t$ according to the Markovian dynamics $P(s_t|a_t, s_{t-1})$ of the environment. This generates a trajectory of states, actions and rewards $\tau = (s_0, a_0, r_0, \ldots)$.

The goal is to maximize the discounted sum of rewards along sampled trajectories

$$J(\theta) = \mathbb{E}_\tau \left[ \sum_{t=0}^\infty \gamma^t r_t \right] = \mathbb{E}_{s \sim \rho^\pi(s), a \sim \pi} \left[ \sum_{t=0}^\infty \gamma^t r_t \right],$$

where $\gamma \in [0, 1)$ is a discount parameter, $\rho^\pi(s) = \sum_{t=0}^\infty \gamma^t P^\pi(s_t = s)$ is the unnormalized discounted state visitation frequency, and $\theta$ parameterizes the policy.

Policy gradient methods differentiate the expected return objective with respect to the policy parameters and apply gradient-based optimization (Sutton & Barto, 1998). The policy gradient can be written as an expectation amenable to Monte Carlo estimation

$$\nabla J(\theta) = \mathbb{E}_{s \sim \rho^\pi(s), a \sim \pi} [Q^\pi(s, a) \nabla \log \pi(a|s)]$$
$$= \mathbb{E}_{s \sim \rho^\pi(s), a \sim \pi} [A^\pi(s, a) \nabla \log \pi(a|s)]$$

where $Q^\pi(s, a) = \mathbb{E}_\tau [\sum_{t=0}^\infty \gamma^t r_t | s_0 = s, a_0 = a]$ is the state-action value function, $V^\pi(s) = \mathbb{E}_{a \sim \pi} [Q^\pi(s, a)]$ is the value function, and $A^\pi(s, a) = Q^\pi(s, a) - V^\pi(s)$ is the advantage function. The equality in the last line follows from the fact that $\mathbb{E}_{a \sim \pi} [\nabla \log \pi(a|s)] = 0$ (Williams, 1992).

In practice, most policy gradient methods (including this paper) use the *undiscounted* state visitation frequencies (i.e., $\gamma = 1$ for $\rho^\pi(s)$), which produces a biased estimator for $\nabla J(\theta)$ and more closely aligns with maximizing average reward Thomas (2014).

We can estimate the gradient with a Monte-Carlo estimator

$$\hat{g} = \hat{A}(s, a, \tau) \nabla_\theta \log \pi_\theta(a|s), \tag{3}$$

where $\tau$ is the rest of the trajectory and $\hat{A}$ is an estimator of the advantage function up to a state-dependent constant (e.g., $\sum_t \gamma^t r_t$). Next, we review common estimators for the advantage function.

### 5.1 ADVANTAGE FUNCTION ESTIMATION

Given a value function estimator, $\hat{V}(s)$, we can form a $k$-step advantage function estimator,

$$\hat{A}^{(k)}(s_t, a_t, \tau_{t+1}) = \sum_{i=0}^{k-1} \gamma^i r_{t+i} + \gamma^k \hat{V}(s_{t+k}) - \hat{V}(s_t),$$

where $k \in \{1, 2, ..., \infty\}$. $\hat{A}^{(\infty)}(s_t, a_t, \tau_{t+1})$ produces an unbiased gradient estimator when used in Eq. 3 regardless of the choice of $\hat{V}(s)$. However, the other estimators ($k < \infty$) produce biased estimates unless $\hat{V}(s) = V^\pi(s)$. Advantage actor critic (A2C and A3C) methods Mnih et al. (2016) and generalized advantage estimators (GAE) Schulman et al. (2015b) use a single or linear combination of $\hat{A}^{(k)}$ estimators as the advantage estimator in $\hat{g}$. In practice, the value function estimator is never perfect, so these methods result in biased gradient estimates. As a result, the hyperparameters that control the combination of $\hat{A}^{(k)}$ must be carefully tuned to balance bias and variance (Schulman et al., 2015b), demonstrating the perils and sensitivity of biased gradient estimators. For the experiments in this paper, unless stated otherwise, we use the GAE estimator. Our focus will be on the additional bias introduced beyond that of GAE.

### 5.2 BASELINES FOR VARIANCE REDUCTION

The policy gradient estimator in Eq. 3 typically suffers from high variance. Control variates are a well-studied technique for reducing variance in Monte Carlo estimators without biasing the estimator (Owen, 2013). They require a correlated function whose expectation we can analytically evaluate or estimate with low variance. Because $\mathbb{E}_{a \sim \pi} [\nabla \log \pi(a|s)] = 0$, any function of the form $\phi(s)\nabla \log \pi(a|s)$ can serve as a control variate, where $\phi(s)$ is commonly referred to as a baseline (Williams, 1992). By using a baseline, the policy gradient estimator becomes

$$\hat{g} = \left( \hat{A}(s, a, \tau) - \phi(s) \right) \nabla \log \pi(a|s),$$

which does not introduce bias.

Several recent methods (Gu et al., 2017b; Thomas & Brunskill, 2017; Grathwohl et al., 2018; Liu et al., 2018; Wu et al., 2018) have extended the approach to state-action-dependent baselines (i.e., $\phi(s, a)$ is a function of the state and the action). With a state-action dependent baseline $\phi(s, a)$, the policy gradient estimator is

$$\hat{g} = \left( \hat{A}(s, a, \tau) - \phi(s, a) \right) \nabla \log \pi(a|s) \\ + \nabla \mathbb{E}_{a \sim \pi} [\phi(s, a)], \tag{4}$$

Now, $\mathbb{E}_{a \sim \pi} [\phi(s, a) \nabla \log \pi(a|s)] \neq 0$ in general, so it must be analytically evaluated or estimated with low variance for the baseline to be effective.

When the action set is discrete, it is straightforward to analytically evaluate the expectation (Gu et al., 2017a; Gruslys et al., 2017; Grathwohl et al., 2018). In the continuous action case, Gu et al. (2017b) set $\phi(s, a)$ to be the first order Taylor expansion of a learned advantage function approximator. Because $\phi(s, a)$ is linear in $a$, the expectation can be analytically computed. Gu et al. (2017a); Liu et al. (2018); Grathwohl et al. (2018) set $\phi(s, a)$ to be a learned function approximator and leverage the reparameterization trick to estimate $\mathbb{E}_{a \sim \pi} [\phi(s, a) \nabla \log \pi(a|s)]$ with low variance when $\pi$ is reparameterizable (Kingma & Welling, 2013; Rezende et al., 2014).

## 6 POLICY GRADIENT VARIANCE DECOMPOSITION

Now, we analyze the variance of the policy gradient estimator with a state-action dependent baseline (Eq. 4). This is an unbiased estimator of $\mathbb{E}_{s, a, \tau} \left[ \hat{A}(s, a, \tau) \nabla \log \pi(a|s) \right]$ for any choice of $\phi$. For theoretical analysis, we assume we can analytically evaluate the expectation over $a$ in the second term because it only depends on $\phi$ and $\pi$, which we can evaluate multiple times without explicitly querying the environment.

The variance of the policy gradient estimator, $\Sigma := \mathrm{Var}_{s, a, \tau}(\hat{g})$ can be decomposed as

$$\Sigma = \mathbb{E}_s \mathrm{Var}_{a, \tau|s} \left( \left( \hat{A}(s, a, \tau) - \phi(s, a) \right) \nabla \log \pi(a|s) \right) \\ + \mathrm{Var}_s \mathbb{E}_{a, \tau|s} \left[ \hat{A}(s, a, \tau) \nabla \log \pi(a|s) \right],$$

where the simplification of the second term is because the baseline does not introduce bias. We can further decompose the first term,

$$\mathbb{E}_s \mathrm{Var}_{a, \tau|s} \left( \left( \hat{A}(s, a, \tau) - \phi(s, a) \right) \nabla \log \pi(a|s) \right) \\ = \mathbb{E}_{s, a} \left[ \mathrm{Var}_{\tau|s, a} \left( \hat{A}(s, a, \tau) \nabla \log \pi(a|s) \right) \right] \\ + \mathbb{E}_s \mathrm{Var}_{a|s} \left( \left( \hat{A}(s, a) - \phi(s, a) \right) \nabla \log \pi(a|s) \right),$$

where $\hat{A}(s,a) = \mathbb{E}_{\tau|s,a}\left[\hat{A}(s,a,\tau)\right]$. Putting the terms together, we arrive at the following:

$$\Sigma = \underbrace{\mathbb{E}_{s,a}\left[\mathrm{Var}_{\tau|s,a}\left(\hat{A}(s,a,\tau)\nabla\log\pi(a|s)\right)\right]}_{\Sigma_\tau}$$
$$+ \underbrace{\mathbb{E}_s\left[\mathrm{Var}_{a|s}\left(\left(\hat{A}(s,a) - \phi(s,a)\right)\nabla\log\pi(a|s)\right)\right]}_{\Sigma_a}$$
$$+ \underbrace{\mathrm{Var}_s\left(\mathbb{E}_{a|s}\left[\hat{A}(s,a)\nabla\log\pi(a|s)\right]\right)}_{\Sigma_s}. \tag{5}$$

Notably, only the second term involves $\phi$, and it is clear that the variance minimizing choice of $\phi(s,a)$ is $\hat{A}(s,a)$. For example, if $\hat{A}(s,a,\tau) = \sum_t \gamma^t r_t$, the discounted return, then the optimal choice of $\phi(s,a)$ is $\hat{Q}(s,a) = \mathbb{E}_{\tau|s,a}\left[\sum_t \gamma_t r_t\right] = Q^\pi(s,a)$, the state-action value function.

The relative magnitudes of the variance terms will determine the effectiveness of the optimal state-action-dependent control variate. The variance in on-policy gradient estimate arises from the fact that we only collect data from a limited number of states $s$, that we only take a single action $a$ in each state, and that we only rollout a single path from there on $\tau$. Intuitively, $\Sigma_\tau$ describes the variance due to sampling a single $\tau$, $\Sigma_a$ describes the variance due to sampling a single $a$, and lastly $\Sigma_s$ describes the variance coming from visiting a limited number of states. The magnitude of these terms depend on task specific parameters and the policy.

## 7 EXPERIMENT DETAILS

### 7.1 Q-PROP EXPERIMENTS

For these experiments, we ran the original Q-Prop code published by the authors at `https://github.com/shaneshixiang/rllabplusplus`. In both of the unbiased and biased variants of Q-Prop, we use the conservative variant of Q-Prop, as is used throughout the experimental section in the original paper. We used the default choices of policy and value functions, learning rates, and other hyperparameters as dictated by the code.

Although Q-Prop and IPG (Gu et al., 2017a) (when $\nu = 0$) claim to be unbiased, the implementations of Q-Prop and IPG apply an adaptive normalization to only some of the estimator terms, which introduces a bias. Practical implementations of policy gradient methods (Mnih & Gregor, 2014; Schulman et al., 2015b; Duan et al., 2016) often normalize the advantage estimate $\hat{A}$, also commonly referred to as the *learning signal*, to unit variance with batch statistics. This effectively serves as an adaptive learning rate heuristic that bounds the gradient variance.

The implementations of Q-Prop and IPG normalize the learning signal $\hat{A}(s,a,\tau) - \phi(s,a)$, but not the bias correction term $\nabla\mathbb{E}_{\pi(a|s)}\left[\phi(s,a)\right]$. Explicitly, the estimator with such a normalization is,

$$\hat{g} = \frac{1}{\hat{\sigma}}\left(\hat{A}(s,a,\tau) - \phi(s,a) - \hat{\mu}\right)\nabla\log\pi(a|s)$$
$$+ \nabla\mathbb{E}_{\pi(a|s)}\left[\phi(s,a)\right],$$

where $\hat{\mu}$ and $\hat{\sigma}$ are batch-based estimates of mean and standard deviation of $\hat{A}(s,a,\tau) - \phi(s,a)$. This deviates from the method presented in the paper and introduces bias. In fact, IPG (Gu et al., 2017a) actually analyzes the bias that would be introduced when the first term has a different weight from the bias correction term, proposing such a weight as a means to trade off bias and variance. However, the weight actually used in the implementation is off by the factor $\hat{\sigma}$, and never one (which corresponds to the unbiased case). This introduces an adaptive bias-variance trade-off that constrains the learning signal variance to 1 by automatically adding bias if necessary.

To generate the TRPO and biased Q-Prop results, we run the code as is. To debias the Q-Prop gradient estimator, we divide the bias correction term $\nabla\mathbb{E}_{\pi(a|s)}\left[\phi(s,a)\right]$ by $\hat{\sigma}$.

## 7.2 STEIN CONTROL VARIATE EXPERIMENTS

For these experiments, we used the original Stein control variate code published by the authors at `https://github.com/DartML/PPO-Stein-Control-Variate`. We use all the default hyperparameters suggested by the authors and test on two continuous control environments, HalfCheetah-v1 and Humanoid-v1 using the OpenAI Gym and Mujoco 1.3.

We test five different sets of algorithms in this experiment: PPO, the Stein control variates algorithm as implemented by (Liu et al., 2018), a variant of the biased Stein algorithm that does not use importance sampling to compute the bias correction term (described below), an unbiased state-dependent Stein baseline, and an unbiased state-action-dependent Stein baseline. All of the learned baselines were trained to minimize the approximation to the variance of the gradient estimator.

To run the first two variants, we run the code as is. In the next variant, we estimate $\nabla E_{a \sim \pi} [\phi(s,a)]$ with an extra sample of $a \sim \pi(a|s)$ instead of importance weighting samples from the current batch (see Eq. 20 in (Liu et al., 2018). For the unbiased baselines, ensure that the policy update steps for the current batch are performed before updating the baselines.

## 7.3 BACKPROPAGATING THROUGH THE VOID

For these experiments, we used the code published by the authors (`https://github.com/wgrathwohl/BackpropThroughTheVoidRL`) with the following modifications: 1) for the state-dependent baseline, we update the value function before computing the policy gradient estimate to remain unbiased, 2) we measure the variance of the policy gradient estimator. In the original code, the authors accidentally measure the variance of a gradient estimator that neither method uses. We note that Grathwohl et al. (2018) recently corrected a bug in the code that caused the LAX method to use a different advantage estimator than the base method. We use this bug fix.

## 8 ESTIMATING VARIANCE TERMS

To empirically evaluate the variance terms, we use an unbiased single sample estimator for each of the three variance terms, which can be repeated to drive the measurement variance to an acceptable level.

First, we draw a batch of data according to $\pi$. Then, we select a random state $s$ from the batch and draw $a \sim \pi$.

To estimate the first term, we draw $\tau, \tau' \sim \tau|s,a$ and the estimator is

$$\left( \hat{A}(s,a,\tau)^2 - \hat{A}(s,a,\tau)\hat{A}(s,a,\tau') \right) \nabla \log \pi(a|s)^2.$$

We estimate the second term in two cases: $\phi(s,a) = 0$ and $\phi(s,a) = \hat{A}(s) := E_{a,\tau|s} \left[ \hat{A}(s,a,\tau) \right]^2$. When $\phi(s,a) = 0$, we draw $a'' \sim \pi$ and $\tau'' \sim \tau|s,a'$ and the estimator is

$$\hat{A}(s,a,\tau)\hat{A}(s,a,\tau')\nabla \log \pi(a|s)^2$$
$$- \hat{A}(s,a,\tau)\hat{A}(s,a'',\tau'')\nabla \log \pi(a|s)\nabla \log \pi(a''|s).$$

When $\phi(s,a) = \hat{A}(s)$, we draw $a_1, a_2 \sim \pi$, $\tau_1 \sim \tau|s,a_1$, and $\tau_2 \sim \tau|s,a_2$ and the estimator is

$$\left( \hat{A}(s,a,\tau) - \hat{A}(s,a_1,\tau_1) \right) \left( \hat{A}(s,a,\tau') - \hat{A}(s,a_2,\tau_2) \right)$$
$$\times \nabla \log \pi(a|s)^2$$
$$- \left( \hat{A}(s,a,\tau) - \hat{A}(s,a_1,\tau_1) \right)$$
$$\times \left( \hat{A}(s,a'',\tau'') - \hat{A}(s,a_2,\tau_2) \right)$$
$$\times \nabla \log \pi(a|s)\nabla \log \pi(a''|s).$$

---

[2]When $\hat{A}(s,a,\tau) = \sum_t \gamma^t r_t$, then $\hat{A}(s) = V^\pi(s)$ a standard choice of a state-dependent control variate.

Finally, the third term is

$$\mathrm{Var}_s\left(E_{a|s}\left[\hat{A}(s,a)\nabla\log\pi(a|s)\right]\right)$$

$$= E_s\left[E_{a|s}\left[\hat{A}(s,a)\nabla\log\pi(a|s)\right]^2\right] - E_{s,a,\tau}[\hat{g}]^2$$

$$\leq E_s\left[E_{a|s}\left[\hat{A}(s,a)\nabla\log\pi(a|s)\right]^2\right]$$

Computing an unbiased estimate of $E_{s,a,\tau}[\hat{g}]^2$ is straightforward, but it turns out to be small relative to all other terms, so we are satisfied with an upper bound. We can estimate the upper bound by

$$\hat{A}(s,a,\tau)\hat{A}(s,a'',\tau'')\nabla\log\pi(a|s)\nabla\log\pi(a''|s).$$

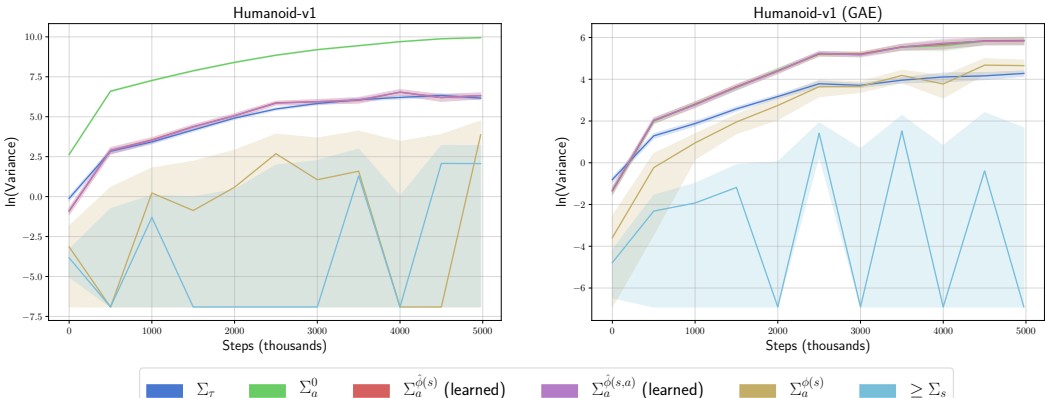

Figure 2: Evaluating the variance terms (Eq. 2) of the gradient estimator when $\hat{A}$ is the discounted return (left) and GAE (right) with various baselines on Humanoid-v1 (See Appendix 3 for results on HalfCheetah-v1). We set $\phi(s) = \mathbb{E}_{a\sim\pi}\left[\hat{A}(s,a)\right]$ and $\phi(s,a) = \hat{A}(s,a)$. The "learned" label in the legend indicates that a function approximator to $\phi$ was used instead of directly using $\phi$. Note that when using $\phi(s,a) = \hat{A}(s,a)$, $\Sigma_a^{\hat{\phi}(s,a)}$ is 0, so is not plotted. Since $\Sigma_s$ is small, we plot an upper bound on $\Sigma_s$. The upper and lower bands indicate two standard errors of the mean. In the left plot, red and purple lines overlap and in the right plot, green, red, and purple lines overlap.

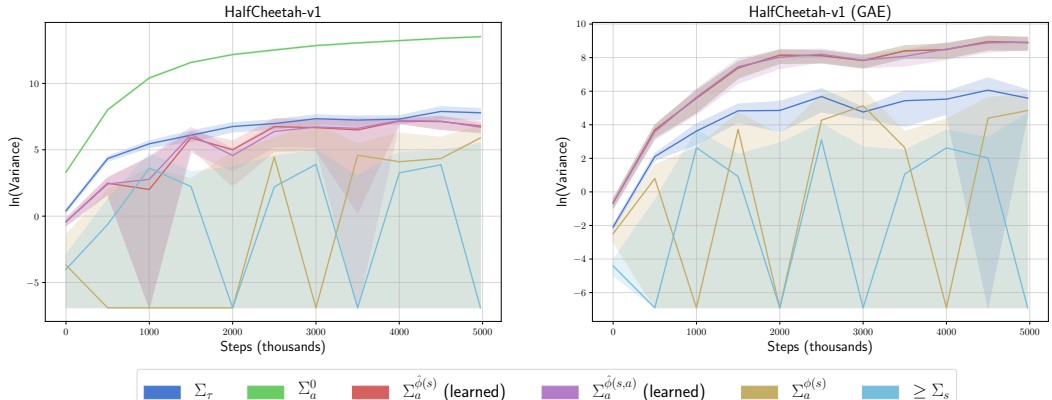

Figure 3: Evaluating the variance terms (Eq. 2) of the gradient estimator when $\hat{A}$ is the discounted return (left) and GAE (right) with various baselines on HalfCheetah-v1. We set $\phi(s) = \mathbb{E}_{a \sim \pi}\left[\hat{A}(s,a)\right]$ and $\phi(s,a) = \hat{A}(s,a)$. The "learned" label in the legend indicates that a function approximator to $\phi$ was used instead of directly using $\phi$. Note that when using $\phi(s,a) = \hat{A}(s,a)$, $\Sigma_a^{\phi(s,a)}$ is 0, so is not plotted. Since $\Sigma_s$ is small, we plot an upper bound on $\Sigma_s$. The upper and lower bands indicate two standard errors of the mean. In the left plot, red and purple lines overlap and in the right plot, green, red, and purple lines overlap.

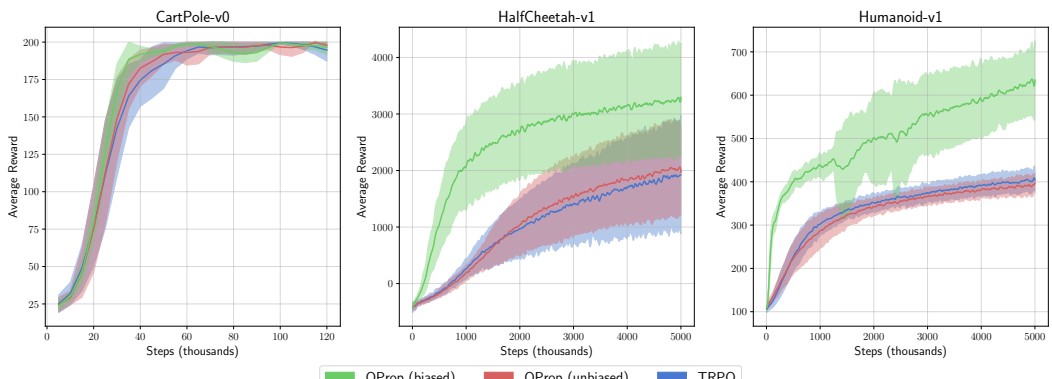

Figure 4: Evaluation of the implementation of Q-Prop from Gu et al. (2017b), an unbiased implementation of Q-Prop that applies the normalization to all terms, and TRPO. We plot average reward across an episode with single standard deviation error intervals capped at the min and max across 10 randomly seeded training runs with the default hyperparameters. The x-axis shows thousands of environment steps, where the batch size across all experiments is 5000. On the continuous control tasks (HalfCheetah and Humanoid), we observe that performance of the unbiased Q-Prop closely follows the TRPO baseline's performance, while the original Q-Prop outperforms both baselines by a wide margin, especially on the Humanoid task. On the discrete control task (CartPole), we observe almost no difference between the three algorithms.

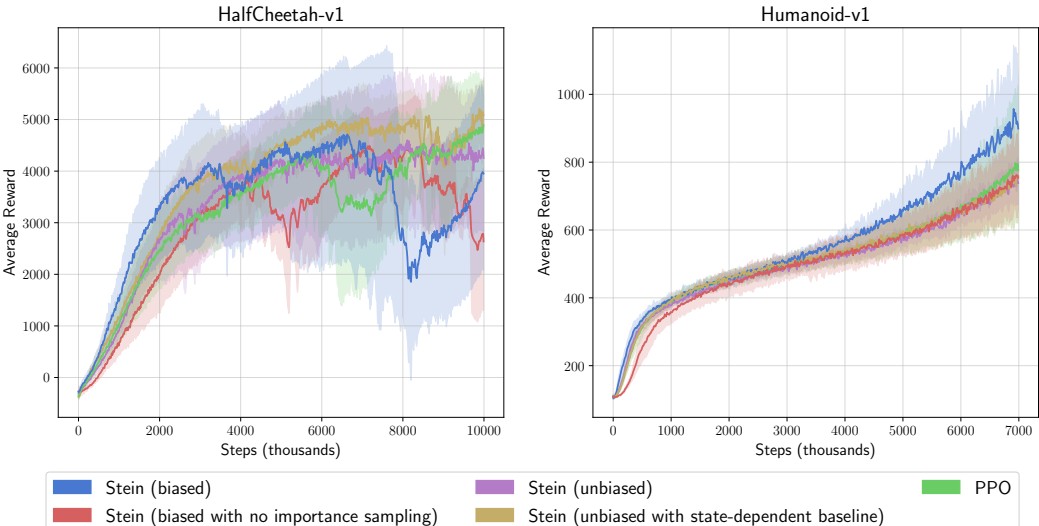

Figure 5: Evaluation of the implementation of the Stein control variate and PPO baseline from (Liu et al., 2018), a biased variant of Stein without importance sampling, an unbiased variant of Stein, and an unbiased variant of Stein using only a state-dependent baseline. Each plot shows average reward across an episode with single standard deviation error intervals capped at the min and max across 5 randomly seeded training runs without any hyperparameter tuning. The x-axis shows thousands of environment steps, where the batch size across all experiments is 10000.

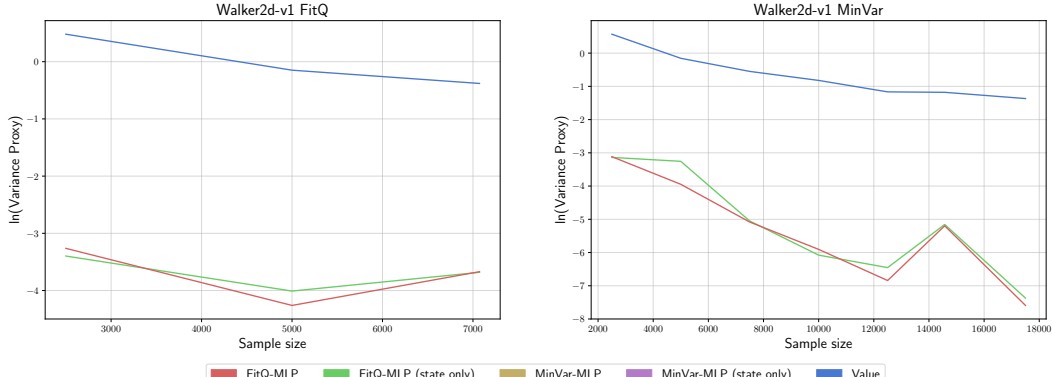

Figure 6: Evaluating an approximation of the variance of the policy gradient estimator with no additional state-dependent baseline (Value), a state-dependent baseline (state only), and a state-action-dependent baseline. FitQ indicates that the baseline was fit by regressing to Monte Carlo returns. MinVar indicates the baseline was fit by minimizing an approximation to the variance of the gradient estimator. We found that the state-dependent and state-action-dependent baselines reduce variance similarly. The large gap between Value and the rest of the methods is due to poor value function fitting. These results were generated by modifying the Stein control variate implementation (Liu et al., 2018).

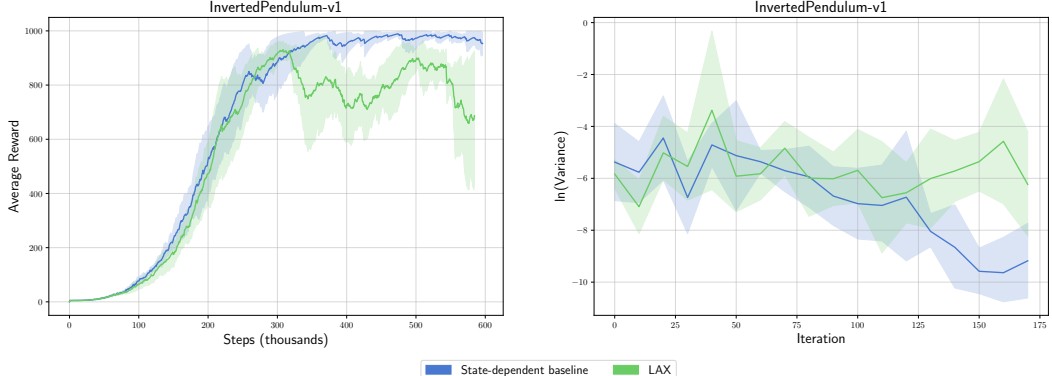

Figure 7: Evaluation of the state-action-dependent baseline (LAX) compared to a state-dependent baseline (Baseline) with the implementation from Grathwohl et al. (2018) on the InvertedPendulum-v1 task. We plot episode reward (left) and log variance of the policy gradient estimator (right) averaged across 5 randomly seeded training runs. The error intervals are a single standard deviation capped at the min and max across the 5 runs. We used a batch size of 2500 steps.

