# OpenReview forum: "The Mirage of Action-Dependent Baselines in Reinforcement Learning"
_ICLR.cc/2018/Workshop — Accept_

### Official Review · AnonReviewer3 · 2018-03-09
**State-action baselines do not help reduce of policy gradients when used for continuous control problems.**

**Rating:** 7
**Confidence:** 1

**Review:**

The paper presents an interesting critique of state-action baselines and how such baselines affect the variance of policy gradient estimators. The authors make two main contributions towards establishing efficacy of such baselines.

 (i) The first contribution is to decompose the variance of policy gradients into three components, $\sigma_t, \sigma_a, \sigma_s$. On continuous control tasks it is observed that $\sigama_t$ >> $\sigma_a$. While using an optimal state-action baseline would improve the \sigma_a term, the dominant $\sigma_t$ term still contributes to large variance.

(ii) When using GAE, the $\sigma_t$ term is reduced. However, the state-action and state based baselines produce similar variance estimates and these variance estimates are larger compared to when using an oracle.

(iii) The second contribution is to point out errors in the open source implementation. The first source of error is that in the implementation of Q-Prop/IPG adaptive normalization is applied to only a few terms which can introduce a bias. Same sample reuse (as done in open source implementations) can lead to biased estimation of policy gradients. After removing the bias in the policy gradients, there is no advantage observed in state-action baselines.

I am not an expert in RL. But with my limited understanding,  I can see that the paper clarifies some important misconceptions. It would have been great, if the authors explained in detail why action/state baselines provide improvement in discrete problems.

---

### Official Review · AnonReviewer1 · 2018-03-10
**Interesting paper that decomposes the sources of variance for RL policy gradients, identifies bugs in prior work, and proposes a simple modification that works**

**Rating:** 8
**Confidence:** 5

**Review:**

The paper is well written, and the key contributions are supported by illustrative experiments. I reviewed the arXiv version of the paper (Feb 27 version); a few minor typos still remain (e.g. eqn in 9.2 needs a superscript \Sigma^a). Interesting paper that decomposes the sources of variance for RL policy gradients, identifies bugs in prior work, and proposes a simple modification that works.

Significance: Good. Isolates bugs in prior art, cleanly motivates fixes. The specific parametrization of horizon-aware value functions (Section 5) can be better motivated -- from a stylized variance formula for LQG to the \hat{V} eqn in Section 5 is a big leap! If the parametrization was argued for in a more principled way, I'm sure the paper becomes much more significant (can become the default way for parametrizing value functions).

Clarity: Good. There were a few expt details that were not clearly listed (e.g. for figure 2/Appendix 10, what was the policy pi? Is the x-axis measuring the same quantity as Fig 1 (task horizon)? Was pi being updated online as we move along the x-axis?)

Quality: Excellent.

---

### Official Review · AnonReviewer2 · 2018-03-10
**An interesting observation but with a limited scope**

**Rating:** 6
**Confidence:** 3

**Review:**

In the context of policy gradient methods, the state-dependent baseline has been used to reduce the variance of the gradient estimate. More recently several papers have suggested the use of state-action-dependent baseline, in order to further reduce the variance. This paper empirically studies the benefit of such a state-action-dependent baseline for various control tasks.
Its main theoretical contribution is the decomposition of the variance to several components, corresponding to the variance coming from (roughly speaking) the randomness of trajectory, a baseline-related term, and the randomness of the advantage-weighted gradient of the log-probability of the policy over states.

The paper empirically shows that for certain benchmarks and a certain architectural choice for NN, the variance term corresponding to the baseline contributes less than the term related to the randomness of the trajectory. Therefore, it does not matter much to use a state-action-dependent vs. state-dependent baselines.

The paper tries to justify the success observed in previous work that used state-action-dependent baselines. It identifies some bugs in each implementation, which explains the apparent superior performance. In other words, the benefit was not due to the use of state-action-dependent baseline.

I believe the paper has an interesting observation, albeit with a quite limited scope. The paper does not show an impossibility result that state-action-dependent baseline cannot help (most likely it actually does for some problems, as mentioned in the Discussion section). It does not characterize when we can expect benefit from the action-dependent baseline and when we cannot. What it shows is that for some specific problems, using state-action-dependent baseline, implemented with a particular choice of architecture, does not help. I would consider this as something that might be good to be known by practitioners.

Also the fact that the authors have identified the bugs in other people’s implementations is commendable, and probably helps improving the available codebase.

So overall, I think this is an alright workshop paper, but its contributions are not major.

---

### Public Comment · ~George_Tucker1 · 2018-02-22
**Erratum**

When preparing the code for release, I found that I had mistakenly claimed that the implementation of Backpropagation Through the Void (Grathwohl et al. 2018) updated the value function before forming the policy gradient estimate. On closer inspection, this is *not* an issue with their implementation, however, they still suffer from two additional issues (see Appendix 7.3 for details). This incorrect claim does not affect the experiments we ran with their code and does not change the conclusions. We have changed the text to reflect this and will upload a new version when revisions are allowed.

Finally, we emphasize that these observations are restricted to the continuous control experiments, and the rest of the experiments in that paper (Grathwohl et al. 2018) use a separate codebase that is unaffected.

---

> ### Public Comment · ~George_Tucker1 · 2018-03-02
> **Revised and extended version posted on arXiv**
>
> We have posted a revised and extended version of the paper on arXiv (https://arxiv.org/abs/1802.10031).

---

### Public Comment · ~George_Tucker1 · 2018-04-25
**Revised version**

Thank you for the helpful feedback. We have substantially revised the paper for clarity and tried to more clearly explain when we would expect a benefit from an action-dependent baseline.

A revised and expanded version can be found here: https://arxiv.org/abs/1802.10031

---

### Decision · Program_Chairs · 2018-03-20
**ICLR 2018 Workshop Acceptance Decision**

**Decision:**

Accept

**Comment:**

Congratulations, your paper was accepted to the ICLR workshop.